# Insights into Zika Virus Pathogenesis and Potential Therapeutic Strategies

**DOI:** 10.3390/biomedicines11123316

**Published:** 2023-12-15

**Authors:** Nohemi Camacho-Concha, María E. Santana-Román, Nilda C. Sánchez, Iván Velasco, Victoria Pando-Robles, Gustavo Pedraza-Alva, Leonor Pérez-Martínez

**Affiliations:** 1Laboratorio de Neuroinmunobiología, Departamento de Medicina Molecular y Bioprocesos, Instituto de Biotecnología, Universidad Nacional Autónoma de México, Cuernavaca 62210, Morelos, Mexico; nohemi.camacho@ibt.unam.mx (N.C.-C.); elizabeth.santana@ibt.unam.mx (M.E.S.-R.); sanchezn@uab.edu (N.C.S.); gustavo.pedraza@ibt.unam.mx (G.P.-A.); 2Instituto de Fisiología Celular-Neurociencias, Universidad Nacional Autónoma de México, Ciudad de México 04510, Mexico; ivelasco@ifc.unam.mx; 3Laboratorio de Reprogramación Celular, Instituto Nacional de Neurología y Neurocirugía “Manuel Velasco Suárez”, Ciudad de México 14269, Mexico; 4Centro de Investigación Sobre Enfermedades Infecciosas, Instituto Nacional de Salud Pública, Cuernavaca 62100, Morelos, Mexico; victoria.pando@insp.mx

**Keywords:** Zika virus, MIA, Musashi, microRNAs, microcephaly, cytokines

## Abstract

Zika virus (ZIKV) has emerged as a significant public health threat, reaching pandemic levels in 2016. Human infection with ZIKV can manifest as either asymptomatic or as an acute illness characterized by symptoms such as fever and headache. Moreover, it has been associated with severe neurological complications in adults, including Guillain–Barre syndrome, and devastating fetal abnormalities, like microcephaly. The primary mode of transmission is through *Aedes* spp. mosquitoes, and with half of the world’s population residing in regions where *Aedes aegypti*, the principal vector, thrives, the reemergence of ZIKV remains a concern. This comprehensive review provides insights into the pathogenesis of ZIKV and highlights the key cellular pathways activated upon ZIKV infection. Additionally, we explore the potential of utilizing microRNAs (miRNAs) and phytocompounds as promising strategies to combat ZIKV infection.

## 1. Introduction

Zika virus (ZIKV) is an emerging mosquito-borne virus. It was first identified in Uganda in 1947, where it was responsible for sporadic human cases; in 2007, the first outbreak occurred on Yap Island, with nearly 75% of the population being infected with ZIKV [1]. During 2013 and 2014, a second epidemic was reported in French Polynesia and the Pacific Islands [2]. However, it was not until 2015 that ZIKV transmission reached pandemic levels, arriving on the American continent. The first reported human case of ZIKV infection in the Americas was in Brazil in May 2015 [2]. Subsequently, different countries reported an increase in the incidence of infection, followed by an unexpected reduction in Zika cases since 2017. To date, a total of 89 countries and territories have reported the circulation of this mosquito-borne virus; however, surveillance remains limited globally [3].

Zika virus (ZIKV) and other arboviruses, such as African swine fever virus, Crimean–Congo hemorrhagic fever virus, dengue virus (DENV), and West Nile virus (WNV), are primarily transmitted via arthropods [4,5]. Notably, these viruses share a common characteristic of potential sexual transmission [6], with ZIKV being the most extensively studied in this regard. Furthermore, it has been documented that ZIKV can also be transmitted through other means, including blood transfusion, saliva, breast milk [7], and vertically from mother to fetus [8].

Of particular interest, ZIKV infection and vertical transmission during pregnancy were associated, through case reports, with microcephaly [9], leading the World Health Organization to declare a Public Health Emergency of International Concern in February 2016 [8].

## 2. Clinical Manifestations of ZIKV Infection

Most human infections with ZIKV are asymptomatic (50–80%) [1]. However, symptomatic ZIKV infection has an incubation period of 3 to 14 days and is characterized as a mild illness, with a duration of up to 1 week [10]. Clinical presentations include an acute febrile illness that is frequently accompanied by headache, joint pain, muscle pain, conjunctivitis, and maculopapular rash, often leading to the confusion of the symptoms with other arboviral infections like dengue or chikungunya [1].

The most severe manifestations of ZIKV infection include Guillain–Barré syndrome (GBS) in adults and congenital Zika virus syndrome (CZS) in newborns [2]. GBS is a severe acute neuropathy characterized by progressive muscle weakness and diminished deep tendon reflexes in weakened limbs, often following a viral or bacterial infection. The highest incidence of GBS following ZIKV infection has been reported through case reports [11,12] and is characterized by lower-extremity weakness, pain, autonomic dysfunction, and facial palsy [13]. A case-control study in French Polynesia reported that 0.25 per 1000 individuals with ZIKV infection develop GBS [14]. However, CZS occurs in newborns exposed to ZIKV during pregnancy. Some characteristics of this syndrome include microcephaly, parenchymal or cerebellar calcifications, ventriculomegaly, central nervous system hypoplasia or atrophy, abnormal visual function, and hearing deficits [15,16]. A prospective study collecting data showed that newborns exposed to ZIKV during pregnancy presented some abnormalities associated with CZS [17].

## 3. Microcephaly Associated with ZIKV

Microcephaly is an unusual condition wherein a baby is born with an abnormally small head, a result of the depletion of the radial glia population, either by cell death, cell cycle arrest, or premature differentiation [18]. Prior to ZIKV infection, the potential causes of microcephaly included infections (e.g., rubella, toxoplasmosis, or cytomegalovirus), maternal malnutrition, drug abuse, genetic factors, or environmental exposures during pregnancy [19]. However, the increase in the number of microcephaly cases and other birth defects reported in northeast Brazil during the first outbreak of ZIKV in South America in October 2015 placed ZIKV infection as a possible cause of microcephaly [20]. Initially, the rate of microcephaly was 48 per 10,000 births in the northeast region of Brazil. Although it has not been repeated in other Brazilian states (5.5–14.5 per 10,000 births in the southeast and center-west, respectively) or in other countries where ZIKV has spread, the peak was 24 times higher than the average occurrence of microcephaly [21,22]. Recently, the U. S. territories and freely associated states collected data from pregnant women with ZIKV infection during pregnancy, and showed that some live-born children had CZS [23].

The early childhood development of children born with congenital microcephaly who are diagnosed with ZIKV infection has been well characterized [24]. The growth and development of such children, between birth and 26 months, have been evaluated, revealing feeding problems, sleep difficulties, severe motor deterioration, cerebral palsy, vision and hearing abnormalities, and seizures [25,26], suggesting that virus infection causes severe neurological impairments during fetal development and in the first two years of life of infected individuals. However, larger studies are needed to understand the full extent of neurological and neuropsychological implications. Although ZIKV infection in pregnant mothers has been widely associated with an increased incidence of microcephaly [20,27], the mechanisms by which the virus causes microcephaly are poorly understood. Nonetheless, several studies have provided some molecular evidence that could explain the association between ZIKV infection and microcephaly.

## 4. ZIKV Interactions with the Host Cell

ZIKV is an enveloped virus that belongs to the *Flaviviridae* family and the genus *Flavivirus*. The ZIKV genome consists of a single-stranded positive RNA of approximately 11 Kb in length, flanked by two untranslated regions with a “cap”-type structure at the 5′ end and a single open reading frame [28]. This genome codes for a polyprotein that is processed via viral and cellular proteases into nonstructural proteins (NS1, NS2A, NS2B, NS3, NS4A, NS4B, and NS5) and structural proteins (prM, C, and E) [28]. Both nonstructural and structural proteins play critical roles in the replication of flaviviruses, the assembly of new viruses, and the evasion of the immune response [29]. ZIKV isolates have been classified into the ancestral African lineage and the emerging Asian lineage [30]. The strains circulating in the Americas belong to the Asian lineage [30,31,32,33].

### 4.1. ZIKV Dissemination Strategy

The strategy for ZIKV dissemination begins in skin fibroblasts, epidermal keratinocytes, and dendritic cells near the site of inoculation, where the virus initially replicates. Subsequently, dendritic cells migrate to lymph nodes and the bloodstream, facilitating the spread of the virus [34,35]. Notably, ZIKV exhibits broad tissue and cell tropism, infecting neural progenitor cells, mature neurons, cord blood cells, several placental cell types, and fetal brain, eye, spleen, and liver cells [36].

ZIKV persists in the genitourinary tract in infected patients (semen, urine, and vaginal secretions) over extended periods and is not commonly observed in flavivirus infections [37,38]. Further studies are required to better understand the interactions of ZIKV with reproductive tissues, how persistent infection modifies female reproductive functions, and the risks of enhancing non-mosquito transmissions.

### 4.2. ZIKV Entry

The cell biology of ZIKV entry remains relatively unexplored. ZIKV’s ability to infect cells is related to the exposure of negatively charged lipids (phosphatidylserine) on the viral envelope surface and the recognition of the viral E protein by host receptors [27]. In this regard, interactions of the viral E protein with the neural cell adhesion molecule (NCAM1) have been described as a potential ZIKV receptor in the infection of astrocytoma cells (U-251 MG) [39]. Additionally, modifications in the viral E protein contribute to better recognition by host receptors. For instance, E glycosylation facilitates the infection of cells expressing C-type lectin dendritic-cell-specific intercellular adhesion molecule-3-grabbing nonintegrin (DC-SIGN, also named CD209), thus contributing to ZIKV pathogenesis [40]. Furthermore, the polyubiquitination of the E protein via the E3-ubiquitin ligase TRIM7 enhances virus attachment and entry into cells expressing the T-cell immunoglobulin mucin receptor 1 (TIM1, also named HAVCR1) [41].

Furthermore, a ZIKV entry mechanism termed “apoptotic mimicry” has been described, where phosphatidylserine found in the viral membrane is recognized by transmembrane receptors of TAM (TYRO3, AXL, and MERTK)—a tyrosine kinase family—and TIM1. These receptors are present in phagocytic cells and differentiate phosphatidylserine from apoptotic cells [42,43]. T-cell immunoglobulin and mucin 3 (TIM3) have been proposed as entry receptors for ZIKV in monocyte-derived dendritic cells (mDCs) [44]. Additionally, tyrosine-protein kinase receptor 3 (TYRO3), CD209, and TIM1 contribute to ZIKV entry in human dermal fibroblasts, epidermal keratinocytes, and immature dendritic cells [35].

In placental cell cultures and chorionic villi explants, susceptibility to ZIKV infection is mediated via AXL, TYRO3, and TIM1. In this model, AXL and TYRO3 expression changes according to cell type, differentiation state, and gestational age, while TIM1 expression remains consistent. This suggests that TIM1 plays a critical role at the uterus-placenta interface and could be a target for the prevention of ZIKV infection [42]. In human pluripotent stem cells (hPSC)-derived microglia, ZIKV infection induced the differential expression of AXL, TYRO3, MERKT, and TIM [45]. Knockout mice for Mertk and Tyro3 were still susceptible to ZIKV and exhibited similar pathogenesis, viral replication, and clinical manifestations compared to WT mice [46]. Nevertheless, the expression patterns of these receptors are limited compared to AXL [47].

AXL is a phosphatidylserine receptor, a member of the TAM family of receptor tyrosine kinases, composed of extracellular, transmembrane, and intracellular domains. Its extracellular structure includes two immunoglobulin (Ig)-like repeats and two fibronectin type III (Fn III)-like repeats, which resemble neural cell adhesion molecules. The Ig motifs are involved in the binding of AXL with its ligand Growth Arrest-Specific Protein 6 (Gas6). Upon ligand binding, the receptor dimerizes, undergoes autophosphorylation on tyrosine residues within its intracellular domain, and subsequently recruits downstream signaling effectors [48]. AXL plays roles in several signaling pathways, including proliferation, apoptosis inhibition, and inflammation modulation [49].

Studies using neutralizing antibodies or small interfering RNA targeting AXL have significantly reduced ZIKV infection in human skin fibroblasts, proposing AXL as a potential entry receptor for ZIKV [35]. In glial cells, AXL is necessary for ZIKV infection. The phosphatidylserine is recognized by GAS6, which serves as a bridge for the interaction between ZIKV and AXL, thereby promoting viral entry. Following this, ZIKV is endocytosed via clathrin-coated vesicles, which are dependent on dynamin-2 function [50]. Also, AXL is widely expressed during neurogenesis, being found in endothelial cells, radial glial cells (RGCs), astrocytes, microglia, and neural stem cells (NSCs) in the human cerebral cortex and human progenitor cells of the fetal retina at 18 gestational weeks (GWSs) [47]. AXL expression becomes denser in microglia and RGC at more advanced stages of development (26 GWS). AXL expression is also conserved in the embryonic cerebral cortex of mice and ferrets, as well as in hPSC-derived cerebral organoids [50]. In line with AXL expression, the retinal cell tropism of ZIKV coincides with the AXL-specific expression pattern in C57BL/6 ZIKV-infected embryos [51], and the inhibition of AXL expression suppresses ZIKV infection in human microglia and astrocytes [50]. However, the genetic ablation of AXL using CRISPR-Cas9 in neural precursor cells (NPCs) and cerebral organoids was insufficient in inhibiting ZIKV infection [52]. Knockout mice for Axl infected with ZIKV exhibited similar viral loads, clinical manifestations, viral distributions, and survival rates compared to WT mice [46,53], suggesting that AXL may not be an indispensable factor for ZIKV infection or that ZIKV infection might involve multiple receptors. Furthermore, the identification of ZIKV interactions with proteins of the caveolar pathway for endocytosis [39] suggests that alternative entry routes into cells may exist. These findings collectively imply that ZIKV entry into cells likely involves multiple receptors, and the differential expression of these receptors is of significance. While not all of these receptors have been discovered, further research is warranted.

### 4.3. Antiviral Response against ZIKV Infection

Following ZIKV entry, the viral genome is recognized in the cytosol via two pattern recognition receptors (PRRs): melanoma-differentiation-associated gene 5 (MDA5) and retinoic-acid-inducible gene 1 (RIG-I). These PRRs can identify viral double-stranded RNA (dsRNA). Subsequently, RIG-I or MDA5 interacts with the mitochondrial antiviral adaptor protein (MAVS) [54]. This interaction recruits multiple signaling components to MAVS, leading to the activation of TANK binding kinase 1 (TBK1), which phosphorylates interferon regulatory factor 3 (IRF3). Phosphorylated IRF3 translocates to the nucleus, initiating the transcription and production of interferon (IFN) α/β (IFN-I) [55]. IFN-I is recognized on the cell surface via IFNα/β receptors (IFNAR), which are associated with Janus kinases (JAKs). These kinases activate the signal transducer and activator of transcription (STAT) proteins that translocate to the nucleus and induce the transcription of interferon stimulated genes (ISGs), which play crucial roles in antiviral defense [56].

During mammalian pregnancy, primary human trophoblasts, which are the barrier cells of the placenta, release IFNλ (IFN-III) to protect both trophoblast and non-trophoblast cells from ZIKV infection [57]. IFN-I and IFN-III treatments have been suggested as therapeutic strategies against ZIKV infection. In ZIKV-infected pregnant mice, these treatments reduced ZIKV infection, protected the fetal brain, and prevented embryonic death by significantly reducing necrosis, inflammation, edema, and hemorrhage. This prevention of fetal miscarriage was achieved through the upregulation of myxovirus resistance protein (MX1) [58]. In the reproductive tract of female mice, IFN-I and IFN-III treatments protect against intravaginal ZIKV infection [59]. Furthermore, during ZIKV infection, transmembrane protein 2 (TMEM2), a member of the interferon-inducible transmembrane protein superfamily, enhances the antiviral response by augmenting the levels of RIG-1, MDA5, pSTAT1, and IFN-β expression [60]. The long isoform of poly(ADP-ribose) polymerase family member 12 (PARP12) mediates the ADP ribosylation of NS1 and NS3, triggering their proteasome-mediated degradation. This process inhibits ZIKV replication and modulates host defense responses [61].

In addition to the interferon response, toll-like receptors (TLRs) play crucial roles in antiviral functions by activating factors related to interferon (IRFs) that, in turn, activate antiviral genes. Early RNA sequencing analysis showed that NPCs infected with ZIKV mounted an antiviral response primarily through TLRs 3/7/8 and 9. Consequently, IRF3 and IRF7 were overactivated, and the canonical NF-κB and STAT inflammation signaling pathways were upregulated, leading to the production of inflammatory cytokines and chemokines (Figure 1) [62]. In human primary astrocytes, dermal fibroblasts, epidermal keratinocytes, and immature dendritic cells, ZIKV induced the transcription of TLR3, and inhibiting it resulted in reduced viral replication [35,63]. However, knockout mice for INFAR 1, IRF3, IRF5, and IRF7 remained susceptible to ZIKV infection and developed neurological disorders but did not succumb to the infection. MAVS and IRF3 mice did not experience weight loss, morbidity, or mortality, suggesting that MAVS and IRF3 might not be required for the antiviral response to ZIKV infection in mice [64].

Microglia, resident cells in the central nervous system (CNS), serve as the primary immune effectors to against pathogens [65]. Interestingly, microglia appear before the onset of neurogenesis and colonize the proliferative zones of the cortex during development [66], where they interact with NPCs [67,68]. During ZIKV infection, microglia are highly susceptible, and it has been suggested that they serve as Trojan horses, contributing to the dissemination of ZIKV in the brain [69]. Additionally, microglia induce cell death in infected developing neurons [70]. They maintain cell viability by phagocytosing apoptotic ZIKV-infected NPCs [45] and viral particles [71], as well as promoting neuroinflammation, primarily through the secretion of cytokines such as CCR5, IL-12, IL-1β, AIF1, IL-6, CCL2, and TNF [45,71,72]. Glial-cell-induced neuroinflammation during ZIKV infection occurs through the activation of inflammatory signaling pathways such as ERK, p38MAPK, NF-κB, and JAK/STAT3. Interestingly, inhibiting p38MAPK after ZIKV infection blocks the inflammatory environment induced by ZIKV and tends to reduce viral titers [73].

Macrophages are another type of immune cell that plays a significant role in recognizing and eliminating invading pathogens. ZIKV infection inhibits macrophage migration inhibitory factor (MIF) expression by disrupting the NF-kB-MIF positive feedback loop, resulting in the prolonged migration of infected macrophages. The migration of infected macrophages can boost ZIKV’s ability to cross physiological barriers and promote virus spread [74]. Like microglia, circulating macrophages, peripheral blood mononuclear cells (PBMCs), and THP1-macrophages infected with ZIKV display a proinflammatory profile mediated via the interaction of ZIKV NS5 with NLRP3 through the NACHT and LRR domains. This interaction promotes the oligomerization of NLRP3 with apoptosis-associated speac-like protein containing caspase recruitment domain (ASC) and subsequently activates caspase 1, leading to the secretion of IL-1β and an inflammatory response in the brain, spleen, liver, and kidneys of mice [75].

### 4.4. Viral Mechanism to Avoid Cell Antiviral Response

There are viral mechanisms designed to counteract cellular antiviral defense. In the context of this, the nonstructural proteins of ZIKV play a crucial role in helping the ZIKV inhibit the host’s innate antiviral immunity, ensuring ZIKV replication and spread. ZIKV utilizes evasion tactics at various levels.

### 4.5. IFN Pathway Suppression

IFN production is initiated upon activation of PRRs such as MDA5/RIG-I. Recent research has revealed that ZIKV NS4A and NS4A2 proteins interfere with the activation of the IFN-β promoter through the MDA5/RIG-I signaling pathway. They disrupt the interaction between RIG-I and MAVS (mitochondrial antiviral signaling protein) and reduce the levels of MAVS, TBK1, and IRF3 proteins [76]. ZIKV NS5 interacts with RIG-I and inhibits the K63-linked polyubiquitination of RIG-I, which reduces the phosphorylation of RIG-I and its nuclear translocation. Consequently, this leads to the suppression of IRF-3 activation [77]. NS1, NS2A, NS2B, and NS4B ZIKV proteins interact within the same pathway to block TBK1 phosphorylation [78,79]. NS4A, in particular, prevents IRF3 phosphorylation, while NS5′s interaction with IRF3 inhibits its nuclear translocation [79]. Intriguingly, the Asian strain of ZIKV is more efficient in suppressing IFN-β activation compared to the African strain, which is primarily due to a single-point mutation (A188V). This mutation enhances the efficiency of the NS1-TBK1 complex and reduces TBK1 phosphorylation [79].

### 4.6. Interference with IFN Signaling

The ZIKV NS3 and NS2B3 complex disrupts the expression of IFN-β by degrading key proteins involved in the antiviral signaling pathway. NS3 degrades MAVS through K48-linked ubiquitination [80], while NS2B3 catalyzes K48-linked polyubiquitination to degrade the Mediator of IFN Regulatory Factor 3 Activation (MITA). Additionally, NS2B3 inhibits K63-linked ubiquitination of MITA, which has a negative effect on the activation of the IFN-β-related antiviral signaling [80].

ZIKV employs NS2B3 to evade the innate immune system downstream of IFN by reducing the JAK/STAT signaling pathway through JAK1 degradation via proteasomal mechanisms [78]. Additionally, ZIKV uses NS5 to promote the degradation of STAT2 in humans through the proteasomal pathway [81]. It also blocks STAT1 phosphorylation, antagonizing type I IFNAR signaling and impairing downstream ISG expression [82].

### 4.7. NLRP3 Inflammasome Activation Suppression

NS1, another ZIKV protein, is involved in avoiding cytokine production by inhibiting NLRP3 inflammasome activation. Within this pathway, NS1 stabilizes caspase 1 through USP8 deubiquitylase. Caspase 1 subsequently interacts with cyclic GMP-AMP synthase (cGAS), a cytosolic DNA sensor that activates the type I IFN pathway. This interaction leads to the cleavage of cGAS and dampens cGAS-STING-mediated IFN production [83]. The NS3 protein impairs NLRP3 activation and IL-1β secretion in primary bone marrow-derived macrophages and mixed glial cell cultures. This mechanism may represent a strategy used by the virus to evade NLRP3 inflammasome-mediated innate immune responses [72].

### 4.8. Interaction with Cellular Proteins

Analyzing ZIKV–human protein–protein interactions has revealed that NS5 also interacts with PAF1C, a chromatin-associated complex that promotes transcriptional elongation. The interaction between NS5 and PAF1C dampens antiviral responses (IFN I) by inhibiting the recruitment of the transcription complex PAF1C to the promoter of antiviral genes [84].

Collectively, these strategies employed by ZIKV help it to evade the host’s innate antiviral immunity, ultimately favoring ZIKV replication.

### 4.9. Maternal Inflammation

Epidemiological studies have consistently demonstrated that inflammatory stimuli, including viral or bacterial infections, experienced by pregnant mothers can trigger a condition known as maternal immune activation (MIA). MIA has emerged as a significant risk factor for the development of neurodevelopmental disorders, including schizophrenia and autism spectrum disorder (ASD) [85,86].

Animal models subjected to MIA during gestation have shed light on its underlying mechanisms. Acute infections in pregnant mothers can lead to transiently elevated levels of cytokines or even initiate autoimmune processes. This, in turn, results in sustained high levels of cytokines throughout the remainder of the pregnancy. Principal among these elevated cytokines found in maternal serum, amniotic fluid, placenta, and fetal brain are TNFα, IL-1β, and IL-6 [87,88,89,90]. These cytokine imbalances during pregnancy can induce significant changes in the fetal environment and impact brain development [91].

In the context of ZIKV infection, studies conducted using human primary culture placental cells have demonstrated that the inflammatory immune imbalances induced by ZIKV can modify lipid metabolism in a way that may lead to placental dysfunction and compromised barrier function [92]. This, in turn, could potentially explain the capacity of ZIKV to spread to the fetus.

In another study involving pregnant rats, those subjected to inflammation induced via lipopolysaccharides (LPSs) during pregnancy exhibited a dysregulation of 3285 genes. These genes were associated with cellular stress, cell death, and nervous system development. Interestingly, the most downregulated genes were those related to nervous system development [87].

Further supporting these findings, experiments involving *Papio anubis* (olive baboons) infected with ZIKV during mid-gestation demonstrated an inflammatory response characterized by increased levels of IL-2, IL-6, IL-7, IL-15, and IL-16 in maternal plasma. The developing fetuses displayed a range of defects, including abnormalities in radial glia, radial glial fibers, disorganized migration of immature neurons to the cortical layers, and signs of astrogliosis (as shown in Figure 2). Additionally, there was an increase in microglia and IL-6 expression in the fetal brain [93].

In-depth analysis of sera from pregnant women infected with ZIKV revealed significant changes in immune profiles. Specifically, there was an induction of 16 inflammatory cytokines and 8 chemokines, along with the repression of 9 inflammatory cytokines and 12 chemokines. Notably, chemokines such as CXCL10, CCL2, and CCL8 were associated with symptomatic ZIKV infection during pregnancy. Moreover, distinct immunoprofiles were detected at different trimesters in ZIKV-infected pregnant women. Of particular interest, the elevated level of CCL2, along with its inverse correlation with CD163, TNFRSF1A, and CCL22 levels, was associated with abnormal births following ZIKV infection [94].

Recent studies conducted on PBMCs from pregnant mothers infected with ZIKV, whether their children had CZS or not, have yielded valuable insights. These studies indicated that T cells from mothers with asymptomatic children tended to exhibit a more inflammatory profile, while T cells from mothers with children diagnosed with CZS tended toward a more cytotoxic profile. This suggests that a mother’s inflammatory profile, whether anti- or pro-inflammatory, plays a significant role in influencing neurological abnormalities in offspring associated with ZIKV infection [95]. It is noteworthy that certain pro-inflammatory cytokines, such as IL-6 and IL-1β, have been shown to reduce adult hippocampal neurogenesis, providing a potential link to the observed neurological effects [96,97].

However, ZIKV congenital infection evades immune response and may persist in the placenta for the duration of pregnancy [98]. Placenta-specific microRNAs (miRNAs) are thought to be a principal marker of viral resistance at the maternal–fetal interface. Placental ZIKV infection disrupts miRNA regulatory networks, leading to altered gene expression of key immunoregulatory pathways, suggesting ZIKV evades normal RNA interference and miRNA regulation mechanisms to persist in the placental niches and render fetal pathogenesis [99].

These findings collectively highlight the complex interplay between maternal immune responses, viral infections, and fetal development, particularly in the context of ZIKV infection during pregnancy.

### 4.10. Signaling Pathways Regulated via ZIKV Associated with Microcephaly

ZIKV infection is associated with a range of fetal brain abnormalities, including disruptions in neural progenitor development, neuronal death, axonal rarefaction, astrogliosis, and microglia activation [24,69,100]. NSCs and NPCs, which are undifferentiated neural cells critical for the developing mammalian nervous system, play a central role in these processes. They have the remarkable capacity to differentiate into various CNS neuronal and glial cell types and possess the ability for self-renewal [101].

Studies using NSCs and NPCs as models have demonstrated that ZIKV can infect different lineages of these cells, leading to apoptosis and the dysregulation of the cell cycle [102]. Interestingly, the presence of the viral E protein in NSCs is sufficient to induce cell cycle arrest, specifically in the G0 phase [103]. This dysregulation of the cell cycle is manifested as inhibited cell proliferation, both in NSCs and NPCs cultured in vitro, as well as in the ventricular zone (VZ) and sub-ventricular zone (SVZ) of the developing brains of embryonic mice (E13.5) [102,103].

Apart from cell cycle dysregulation and the inhibition of proliferation, apoptosis is another process affected via ZIKV infection. While ZIKV can inhibit apoptosis in fibroblasts to ensure viral replication, studies involving hPSCs have revealed that ZIKV infection induces an increase in caspase 3 activity [62,104]. Additionally, in human NSC derived from the H9 cell line, infection with ZIKV strains, both Asian (PVRABC59) and African (MR766), activates cellular responses to damage. The African strain leads to increased phosphorylation of histone H2AX and the activation of caspase 3, while the Asian strain promotes the phosphorylation of p53, p21, and PUMA, ultimately leading to cell cycle arrest [105]. Interestingly, in the human H9 cell line, ZIKV infection leads to an upregulation of p65 and IRF3 in response to viral infection. However, this upregulation of NF-κB and IRF3 has been associated with growth arrest and reduced differentiation in NPCs [106].

ZIKV infection also induces alterations in the differentiation process of neural cells. For example, in the VZ of ZIKV-infected brains at E16.5, the virus inhibits the transition from radial glial cells (marked by Pax6 expression) to intermediate progenitor cells (marked by Tbr2 expression) [102]. In human NPCs (hNPCs), ZIKV infection induces early differentiation, characterized by an increase in the expression of pro-neural genes, such as PTN, ROBO2, SPOCK1, and DCX [103]. Interestingly, although seemingly contradictory, reports indicate that cell differentiation occurs in a specific manner. In mouse NSCs, ZIKV infection reduces the expression of marker genes associated with neuronal and oligodendrocyte progenitors while increasing the expression of markers associated with astrocyte progenitors [107]. Similarly, in ZIKV-infected mouse embryos at E15.5, impairments in gliogenesis are observed due to reduced cell proliferation and differentiation of oligodendrocyte precursor cells, resulting in less myelination and neuronal loss [108]. Studies on *Macaca nemestrina* infected with ZIKV have revealed increased astrogliosis, accompanied by a reduction in NPCs, a loss of non-cortical volume in the fetal brain, and perturbations in neuron maturation in the hippocampus and cerebral cortex [109].

Additionally, ZIKV infection impacts the integrity of cellular adherent junctions (AJs) in the developing mammalian brain. Researchers have observed that the viral protein NS2A can interact with multiple AJ complexes, leading to their destabilization. This results in poor AJ formation and abnormal scaffolding of radial glial fibers in mouse embryonic cortices [110].

Transcriptomic analyses of ZIKV-infected hNPCs have revealed that ZIKV infection induces changes in the expression of alternative splicing, gene isoform composition, and long non-coding RNAs (lncRNAs) implicated in processes such as DNA replication, cell cycle regulation, apoptosis, cell death, RNA processing, immune responses, and neuron development [111]. This multifaceted impact on gene expression and regulatory processes underscores the complexity of ZIKV-induced developmental abnormalities in the brain.

### 4.11. Unfolded Protein Response (UPR) and Autophagy during Zika Infection

During embryonic development, endoplasmic reticulum (ER) stress and the UPR pathway can lead to a decrease in neurogenesis, ultimately resulting in microcephaly [112]. The UPR pathway is modulated via three receptors anchored to the ER membrane: endoplasmic reticulum kinase similar to PKR (PERK), inositol-requiring enzyme 1 alpha (IRE1α), and activating factor of transcription 6 (ATF6) [113]. In the context of infection with RNA viruses like ZIKV, there is a remodeling of the ER membrane, inducing stress and subsequently activating the UPR [114]. Studies involving human fetuses at 22 weeks of gestation (22GW), human NSCs, and mouse embryos infected with ZIKV provide evidence that ZIKV infection triggers ER stress and the UPR in the cerebral cortex [94]. This ER stress is associated with a decrease in the proliferation of cortical progenitors and an increase in the apoptosis of mature neurons in the cerebral cortex [115]. In cultures of human cortical neurons derived from induced PSCs infected with ZIKV, the UPR is activated through ATF4 and IRE1-XBP1. Additionally, CHOP is overexpressed in the cortical plate, VZ, and SVZ, which triggers apoptosis in these cells [113,114].

In immortalized neuroblastoma and astrocytoma cell lines derived from AG6 mice, ZIKV infection has been shown to activate the pIRE1-XBP1 and ATF6 pathways of the unfolded protein response [116]. The activation of UPR via ZIKV disrupts the balance between direct and indirect neurogenesis, resulting in fewer resident neurons in the upper layer of the cerebral cortex [93,94]. Furthermore, in human embryonic astrocytes infected with ZIKV, the miRNA-30e-5p, -19b-3p, and -17-5p are upregulated, leading to the induction of UPR and the subsequent activation of the apoptosis markers CHOP and GADD34 [117]. Another cellular protein that undergoes changes during ZIKV infection is the Hsp70 chaperone, which is recruited to the replication complex to facilitate proper ZIKV replication. Additionally, Hsp70 plays a crucial role in maintaining capsid protein stability and promoting the assembly of ZIKV infectious particles [118].

Transcriptomic analyses of ZIKV-infected hNPCs have revealed alterations in the Akt-mTOR pathways [119]. Interestingly, NPCs from ZIKV-exposed babies are more permissive to ZIKV replication compared to NPCs from unaffected babies, and this difference is attributed to genes related to the Wnt and mTOR pathways. The activation of the mTOR pathway inhibits the release of viral particles from NPCs, while the repression of the mTOR pathway promotes viral particle release [120]. In hfNSCs, the co-expression of ZIKV NS4A and NS4B suppresses Akt phosphorylation, leading to reduced activation of mTOR and the induction of autophagy. This favors viral replication in hfNSCs while inhibiting neurogenesis [121]. Therefore, inhibiting autophagy with specific drugs has been shown to decrease ZIKV viral progeny in fibroblast cells [35].

### 4.12. miRNAs and Musashi 1 Interaction

ZIKV has been shown to induce significant changes in RNA metabolism, resulting in alterations in lncRNAs, the deregulation of alternative splicing [111], and changes in miRNA expression [103]. For instance, ZIKV infection leads to an increase in miR-9 expression in mice embryos at E12.5, mimicking the microcephaly phenotype. This upregulation of miR-9 is associated with the downregulation of glial-cell-derived neurotrophic factor (GDNF), a survival factor critical for NPCs and immature neurons [122]. Moreover, miRNAs, such as miR-1273g-3p and miR-204-3p, are upregulated in response to ZIKV infection, and these miRNAs target genes like PAX3 and NOTCH [103]. NOTCH plays a crucial role in expanding human cortical progenitors by promoting the increased proliferation of RGCs in the cortex and delaying their differentiation into neurons [123,124].

Cell cycle dysregulation is another consequence of altered miRNA expression during ZIKV infection. When hNSCs are infected with ZIKV, miR-124 levels increase. Elevated miR-124 levels in ZIKV-infected neurospheres result in reduced neurosphere sizes due to defects in cell proliferation. miR-124 accomplishes this by targeting the mRNA of the transferrin receptor (TFRC), leading to reduced levels of the transcription factor FOXM1, which plays a pivotal role in cell cycle progression [125].

Interestingly, in cortical neurons from C57BL/6J mice, miRNA-124 is downregulated, resulting in the increased expression of its target genes [126]. In human embryonic astrocytes infected with ZIKV, miRNA-17-5p is upregulated. This miRNA is well known for its role in inducing the UPR through Hsp70 [112,117].

Furthermore, in a cellular model of neuroblastoma (SH-SY5Y) and post-mortem brains from individuals with CZS, ZIKV infection induces the overexpression of miRNA-145 and miRNA-148. In silico analysis suggests that these miRNAs are related to processes such as CNS function, cellular migration, and adhesion. [127]. However, the specific functions of these miRNAs in the context of ZIKV infection during CNS development have not been extensively studied.

Musashi 1 (MSI1) is a 3′ untranslated region (3′UTR) RNA-binding protein that plays a role in post-transcriptional gene regulation. MSI1 is associated with maintaining stem cell self-renewal and neurogenesis [128]. Knocking out MSI1 in mice results in abnormal brain development [129]. Additionally, MSI1 represses the translation of mRNAs encoding proteins involved in neurogenesis and the cell cycle, such as Numb and p21 (an endocytic adaptor protein and cyclin-dependent kinase inhibitor 1A, respectively) [130]. Bioinformatics approaches have identified Musashi binding elements (MBEs) in the 3′UTR of ZIKV. These MBEs are predominantly found in unpaired and single-stranded structural contexts [131]. In vitro and in vivo studies have shown that neuronal MSI1 interacts with the ZIKV genome, facilitating viral replication [132]. Recently, an interaction between MSI1 and ZIKV has been demonstrated in the AGAA tetraloop of xrRNA2 (Xrn1-resistant RNA) within the 3′UTR of ZIKV [133]. Interestingly, MSI1 can be negatively regulated at the translational level via miRNAs, such as miR-23a and miR-125b in NSC/NPCs [134], miR-137 in NSCs, and miR-330-3p in gastric cancer cells [135,136]. Target prediction programs have identified potential binding sites for miR-125b, miR-137, miR-144, miR-185, and miR-342-3p within the MSI1 3′UTR [137]. Intriguingly, ZIKV downregulates miRNAs that repress MSI1 mRNA, including miRNA-125b, -137, -144, -342-3p, -330-3p, and -23a [125,126,127].

## 5. Therapeutic Strategies versus Zika Virus

### 5.1. ZIKV Vaccines Development

Various vaccine candidates for ZIKV are currently undergoing phase I/II clinical trials. These candidates encompass a range of approaches, including inactivated whole viruses, recombinant measles virus vector-based vaccines, DNA and mRNA vaccines, E protein subunit vaccines, and subviral particles, as well as a mosquito salivary peptide vaccine [138,139].

One of the key challenges in ZIKV vaccine development is determining whether it will be feasible to create an immunogenic and safe vaccine. Given the relatively low variation among ZIKV strains, with approximately 94% amino acid identity across the viral genome, and the absence of different genotypes or serotypes, it is plausible that an effective vaccine targeting one strain could confer broad protection against all circulating ZIKV strains. Furthermore, since ZIKV outbreaks often occur in regions with high seroprevalence rates for DENV infection and previous vaccination with yellow fever virus, a significant portion of potential ZIKV vaccine candidates may induce preexisting cross-reactive antibodies resulting from natural or vaccine-induced flavivirus immunity. Preexisting immunity can influence ZIKV vaccine responses in several ways. It may boost cross-reactive immunity, providing protection against ZIKV. Alternatively, it could enhance cross-reactive immunity while potentially reducing protective ZIKV-specific responses, a phenomenon known as “original antigenic sin”. There’s also the possibility of neutralizing live-attenuated ZIKV without significantly impacting cross-reactive immunity, which is referred to as sterilizing immunity [140]. However, it is essential to note that the preexisting antibody response induced by flaviviruses can enhance disease severity upon ZIKV infection through a phenomenon known as antibody-dependent enhancement (ADE) [141]. ADE is characterized by non-protective cross-reactive memory antibodies that may even enhance the infection and clinical manifestations [142]. Studies have shown that convalescent plasma from individuals infected with DENV and WNV can enhance ZIKV infection in vitro, which is mediated through immunoglobulin G engagement of FcγRIIA receptors. In in vivo models, this resulted in high viremia and increased mortality in a dose-dependent manner of convalescent plasma [143]. Understanding how the antibody response to flavivirus infections contributes to protection versus pathogenesis is crucial in the development of a ZIKV vaccine.

### 5.2. Antivirals

Currently, there are no approved antiviral strategies for combatting ZIKV infection. However, a screening assay of FDA-approved drugs conducted in various human cell lines, including HuH-7, HeLa, JEG3, and NSCs, as well as primary human amnion cells infected with ZIKV, has identified 20 compounds that exhibit a significant reduction in ZIKV infection in vitro. Notable drugs among these include ivermectin, mefloquine hydrochloride, bortezomib, daptomycin, and mycophenolic acid [144].

Furthermore, there are candidate molecules aimed at inhibiting ZIKV replication by targeting specific ZIKV proteins or host factors essential for ZIKV propagation. In this context, the SEC61 translocon, a crucial host factor for ZIKV replication, has been successfully targeted by co-translational translocation inhibitors (cotransins) CT8 and PS3061, resulting in the inhibition of ZIKV virion production and replication [84]. Another promising candidate is AR-12 (OSU-03012), a celecoxib derivative and kinase inhibitor that downregulates the Akt pathway. Celecoxib has demonstrated the ability to inhibit ZIKV in vitro infection and improve the survival rates of individuals deficient in type I interferon receptor (A129) [145]. Furthermore, Hsp70 inhibitors, namely, JG40 and JG345, reduce ZIKV particle production in trophoblasts and hNSCs, and provide protection to IFNR-knockout mice against lethal ZIKV infection [118].

Metformin (MET), an FDA-approved drug, has exhibited inhibitory effects on ZIKV infection in glioblastoma and hepatic cell lines. However, its most significant impact has been observed in inhibiting DENV and yellow fever virus infections, rather than ZIKV. MET treatments hinder the formation of the replicative complex by reducing cellular cholesterol synthesis through the inhibition of 3-hydroxy-3-methylglutaryl-coenzyme A reductase (HMGCR) activity [146].

Chemical compounds also offer the potential for inhibiting ZIKV replication by targeting viral proteins essential for the virus’s life cycle. Sofosbuvir, a class B FDA-approved antiviral used against hepatitis C virus, has been demonstrated to inhibit ZIKV RNA polymerase activity in various cell lines, including Huh-7, SH-SY5Y cells, NSCs, and brain organoids [147]. Ouabain, a cardiotonic steroid, disrupts ZIKV genome replication by binding to regions associated with ATP hydrolysis (NTPase domain), RNA binding domain, and RNA-dependent RNA polymerase (RdRp) domain of NS5 [148].

Furthermore, high-throughput virtual screening studies have facilitated drug discovery, with several commercial compounds, such as CD11575, CD03173, HTS04601, HTS03171, HTS07252, JFD01698, and KM10383, emerging as potential inhibitors of NS2B/NS3 ZIKV proteins [149]. Another chemical compound, (2E)-N-benzyl-3-(4-butoxyphenyl) prop-2-enamide (SBI-0090799), inhibits ZIKV infection by preventing the formation of membranous compartments in the ER, where ZIKV RNA replication occurs, through the suppression of NS4A activity [150]. At the endosomal–lysosomal compartment level, bafilomycin A1, a V-ATPase inhibitor, holds potential as an antiviral target due to its ability to reduce ZIKV release by interfering with viral maturation [151].

The field of antivirals presents a promising avenue for combatting ZIKV infection, with strategies targeting multiple steps in the virus replication cycle or host proteins essential for the virus’s survival. However, it is crucial to exercise caution when inhibiting host proteins, as they often play intricate roles in signaling pathways. Additionally, the proposed compounds should be evaluated in in vivo models.

### 5.3. Phytocompounds

Phytocompounds are emerging as a promising strategy for controlling ZIKV infection. In vitro screening assays have identified several plant extracts, such as lycorine, pretazettine, narciclasine, and narciclasine-4 O-β-D-xylopyranoside, that exhibit antiviral activity against ZIKV [152]. 6-deoxyglucose-diphyllin (DGP), a bioactive phytoconstituent molecule derived from the medicinal plant *Justicia gendarussa*, has demonstrated the ability to block ZIKV infection in cultured cells. Importantly, it also prevented ZIKV-induced mortality in a mouse model lacking IFNAR. DGP achieves this by preventing the acidification of endosomal and lysosomal compartments, inhibiting ZIKV fusion and thereby preventing the delivery of viral RNA into the cytosol of the target cell [153]. Tannic acid, found in *Terminalia arjuna*, interacts with the ZIKV E protein, suppressing E dimerization and membrane fusion with host cells. Notably, tannic acid exhibits a high affinity for the ZIKV E protein and a much less significant affinity for host cells [154]. Another molecule, epigallocatechin gallate (EGCG), isolated from green tea, also interacts with the ZIKV E protein, inhibiting ZIKV entry into host cells [155].

In silico approaches have suggested potential antiviral activity by targeting interactions with the ZIKV NSB protein. Compounds like ECGC-7-oa-glucopyranoside, isoquercetin, rutin, and ECGC-4-oa-glucopyranoside can interact with the active site of the NS2B/3 protease complex, which is crucial for ZIKV replication [156]. Naringenin, a citrus flavanone, has been found to inhibit ZIKV infection in glioblastoma cell lines and dendritic cells, particularly against both Asian and African lineages of ZIKV. Computational analysis suggests that this inhibition occurs through binding to the NS2B/3 protease domain, an essential component of virus assembly [157]. Similarly, EGCG can block viral replication by interacting with the ATPase site in the NS3 viral helicase [158].

Natural compounds from plants are being investigated as modulators of miRNA expression. For example, in breast cancer, compounds like curcumin, resveratrol, isoflavone, EGCG, I3C, and diindolylmethane have been identified as potent agents capable of modulating miRNA expression [159]. Polyphenol-enriched fractions from blueberries and berberine can inhibit the expression of miR-125b [160,161]. Conversely, *Olea europaea* leaf extract induces the upregulation of miR-137 [162], and American ginseng promotes the upregulation of miR-144 [163]. These compounds may hold promise in modulating miRNA expression in the context of ZIKV infection and CNS development.

Natural products offer a rich source of potential drugs or drug-like molecules for treating viral infections. Additionally, chemical modifications of these plant extracts can lead to the development of new derivatives that may be optimized for improved bioavailability, oral absorption, reduced toxicity, and increased antiviral activity. Given that the major complications of ZIKV infections are associated with pregnant women, it is crucial to identify molecules that are safe and have minimal or no side effects.

## 6. Conclusions

Prenatal infection with ZIKV has been associated with an elevated incidence of CZS in newborns. In addition, retrospective studies have revealed various neurological impairments in children under two years old. These observations have prompted extensive research efforts to uncover the underlying mechanisms. Nonetheless, the precise mechanisms by which the virus induces microcephaly and neurological damage remain a subject of investigation. Studies employing cellular, animal, and human models have shed light on ZIKV pathogenesis, revealing its broad tissue and cell tropism and its capacity to cause severe end-organ diseases, including placental and congenital infections.

Recent in vitro and in vivo investigations have provided molecular insights into the association between ZIKV infection, microcephaly, and neurological damage. Additionally, studies have explored the immune response to Zika infection and how the virus evades immune defenses. Nonetheless, numerous unanswered questions persist in understanding the key aspects of ZIKV infection, such as the primary viral antigens, the mechanisms of virulence, host restriction, immune evasion, and the potential for ADE in ZIKV pathogenesis. Furthermore, the long-term consequences of congenital ZIKV infection in humans require further exploration.

Efforts are ongoing to develop effective preventive and therapeutic vaccines against Zika infection. Currently, there is growing interest in the field of metabolites as regulators of miRNA expression. It is proposed that ZIKV infections could be addressed by utilizing phytochemical compounds with the potential to induce antiviral miRNAs or target host proteins necessary for ZIKV replication.

In summary, this review emphasizes the significance of understanding ZIKV pathogenesis, especially its association with microcephaly and neurological damage. It underscores the current knowledge gaps and the importance of ongoing research efforts aimed at elucidating the complex interactions between the virus and the host. Ultimately, the goal is to develop effective interventions to combat ZIKV infection and its associated complications.

## Figures and Tables

**Figure 1 biomedicines-11-03316-f001:**
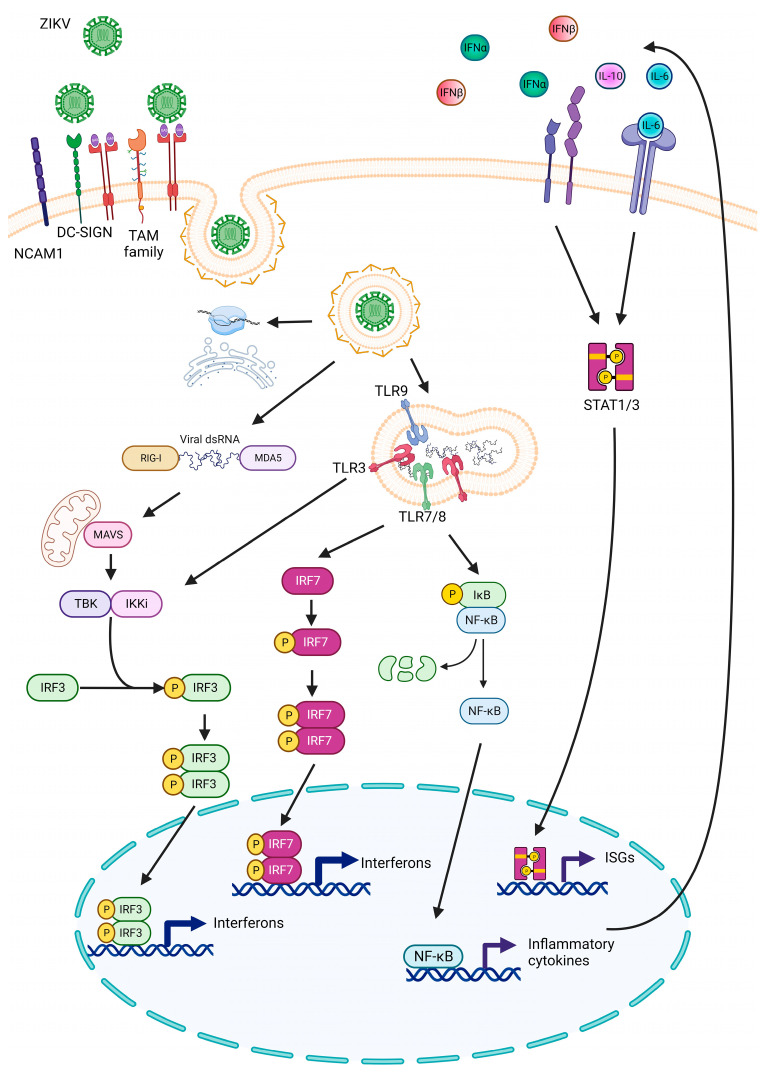
Representative scheme of the main signaling pathways activated against ZIKV infection. ZIKV’s ability to infect cells is related to the presence of phosphatidylserine on the viral envelope protein’s surface, which is recognized via NCAM1, DC-SIGN, and the transmembrane phosphatidylserine receptor of the TAM family. Inside the cell, the viral genome is recognized via two PRRs: MDA5 and RIG-1. These interact with MAVS, leading to the activation of the IRF3 pathway, inducing the transcription and production of IFNα/β. Additionally, the antiviral response through TLRs 3, 7, 8, and 9 is induced. Consequently, IRF3 and IRF7 are overactivated, and canonical NF-κB and STAT inflammation signaling pathways are upregulated, resulting in the production of inflammatory cytokines and chemokines. ISGs: interferon stimulated genes. Created with BioRender.com> (accessed on 20 September 2023).

**Figure 2 biomedicines-11-03316-f002:**
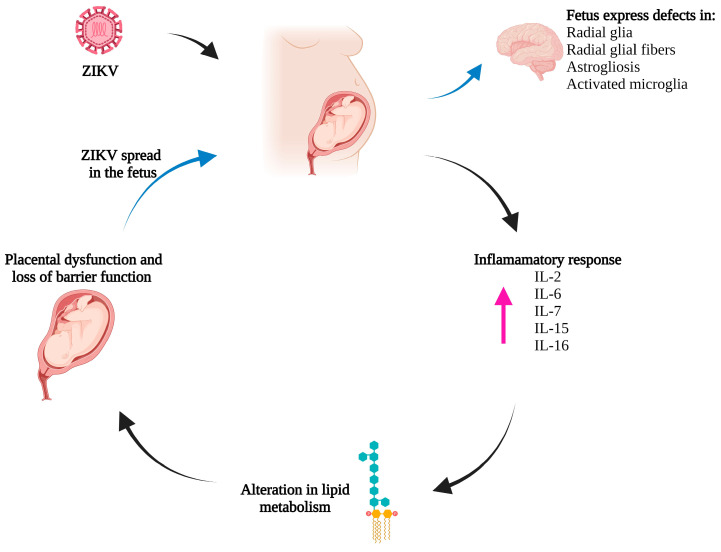
Schematic representation of maternal inflammation induced by ZIKV as a risk factor for neurodevelopmental disorders. Acute infection in pregnant mothers increases cytokine levels or may even initiate autoimmune processes, resulting in sustained high levels of cytokines (indicated by the pink arrow) during pregnancy. Cytokine imbalances during pregnancy induce changes in the fetal environment and can modify lipid metabolism, leading to placental dysfunction and compromising the barrier function (indicated by the black arrows). This placental dysfunction and compromised barrier function could explain ZIKV’s ability to spread to the fetus and induce damage in nervous system development (indicated by the blue arrows). Created with BioRender.com (accessed on 21 October 2023).

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
