# Peer review of "Insights into Zika Virus Pathogenesis and Potential Therapeutic Strategies"

_biomedicines, 2023, doi:10.3390/biomedicines11123316_

Round 1

Reviewer 1 Report

Comments and Suggestions for Authors

The manuscript by Camacho-Concha and colleagues is generally well written and nicely summarizes a great deal of the literature on pathogenesis of Zika virus infection.  There have been several other comprehensive reviews of this topic, most recently an Annual Review, but I think there is reason to think this manuscript will be of value to a somewhat different audience.  Overall, I found this manuscript to be very well organized and do not have major criticisms or suggestions to make, but offer the following rather minor comments:

- Congenital Zika syndrome: the abbreviation CZS is first used on line 73 but not defined until line 312.

- AXL: perhaps define this as a PS receptor

- Line 104+: this sentence is quite awkward and should be rewritten e.g. “infected patients (and present in semen, urine, and vaginal secretions)” .. and not commonly observed in ..

- Line 175: the work cited was in a mouse model – suggest changing “dams” to “mice” to avoid any misinterpretation. The same comment applies to the sentence beginning on line 179 (reference 46)

- Line 228: suggest “Strategies to avoid cell antiviral response” -> Viral mechanisms to avoid cellular antiviral responses.”

- Line 473: may possess -> may induce (vaccines do not possess antibodies)

- Line 505: has demonstrated its ability to inhibit -> has been demonstrated to inhibit

- Line 563: I think “Prenatal infection” would be more accurate than “Congenital infection”

- Line 565: “severe neurological impairments in infected individuals during the first two years of life” implies that they recovery after the first two years; suggest re-wording this sentence.

- General comment on the section for antivirals: it might be good to indicate that all of these in vitro studies need to be validated in some type of in vivo system (e.g. at least mice).

Comments on the Quality of English Language

Overall good - minor editing suggested.

Reviewer 2 Report

Comments and Suggestions for Authors

This review article describes the state of the art regarding Zika virus infection, with emphasis on viral entry and potential treatments. This is not a bad review, but it requires extensive actualization and more attention to detail regarding grammar and use of abbreviations. Finally, there are specific sections that are underdeveloped. However, successfully addressing these points will make this a good entry point for newcomers to the field and a good reference for experts.

MAJOR ISSUES

CZS is introduced in page 7, but it appears earlier in the text. Please describe acronyms and abbreviations the first time they appear.

What is AXL?

Discuss the recently described role of NCAM1 (CD56) in ZIKV entry in neuronal and blood cells.

Describe briefly Guillan-Barré syndrome.  What is the % of ZIKV-infected individuals that may develop GBS?

Please describe in more detail the possible increase in severity by previous exposure to DENV infection (DENV-ZIKV crosstalk) explained in the vaccination section, as well as the possible antibody-dependent enhancement of ZIKV infection, similar to DENV.

Please note that other arbovirus can be transmitted via sexual contact, albeit it is a less efficient process (PMC7552039). This should be discussed.

MINOR COMMENTS

Line 62, replace “Before” with “Prior to”

Line 71, replace “mean” with “average”.

Line 105, replace “sperm” with “semen”.

Line 352, replace “resulting” with “leading to”

Line 377, lead with “Transcriptomic analysis of ZIKV-infected hNPCs have revealed that ZIKV infection induces…”

Comments on the Quality of English Language

English is OK, and only minor changes are required.

Round 2

Reviewer 2 Report

Comments and Suggestions for Authors

The authors have addressed my concerns, and the paper is now acceptable.